# Morphological characteristics and genome-wide association analysis among local *Andrographis paniculata* from Thailand under controlled environment in plant factory

Praderm Wanichananan, Supattana Janta, Suchalee Sueachuen, Tanawut Chiangklang, Kriengkrai Mosaleeyanon, Siripar Korinsak, Clive Terence Darwell, Panita Chutimanukul *

National Center for Genetic Engineering and Biotechnology (BIOTEC), National Science and Technology Development Agency, Klong Luang, Thailand

* panita.chu@biotec.or.th

## Abstract

*Andrographis paniculata* Wall. ex Nees (*A. paniculata*) is a medicinal plant widely used in Southeast Asian traditional medicine. Plant factories with artificial lighting (PFAL) provide controlled environments for optimizing plant growth and quality. However, the variability in biomass and bioactive compound production among *A. paniculata* varieties cultivated in PFAL is not well understood. This study investigated ten locals of *A. paniculata* accessions to assess their growth characteristics and andrographolide (AP1) content in a PFAL system using hydroponic cultivation. Among the accessions, the TTT cultivar showed significantly higher stem height, plant width, above-ground biomass yield, leaf number, and AP1 content compared to others. Phylogenetic analyses based on SNP markers revealed that TTT is morphologically distinct but genetically similar to CR, RB, PL, and PC accessions. A genome-wide association study (GWAS) identified two significant SNP regions on chromosome 9 associated with yield and AP1 content. These findings highlight the potential of TTT as a high-quality cultivar for pharmaceutical use and provide insights into key genes that could be targeted for breeding programs to improve *Andrographis* production in PFAL systems.

## Introduction

*Andrographis paniculata* (Acanthaceae; Burm. F.) Wall. ex Nees is an annual herb that grows to a height of 30–110 cm. It has a long, thin stem that is typically green in color and slightly branched. The leaves are simple, opposite, and lanceolate in shape, with a serrated margin and a smooth surface. They are typically 2–12 cm long and 1–3 cm wide [1]. The flowers of the *A. paniculata* plant are small, white or greenish-white, and arranged in spikes at the top of the stem. The spikes are

**Data availability statement:** All relevant data are within the paper and its Supporting Information files.

**Funding:** This research was funded by the support of National Science and Technology Development Agency, Thailand (P2351505) and Thailand Basic Research Fund: fiscal year 2023 with Contract no. 4709540.

**Competing interests:** The authors have declared that no competing interests exist.

5–15 cm long and contain numerous individual flowers that bloom in succession from the bottom to the top of the spike. The fruit of the *A. paniculata* plant is a small capsule containing one to two seeds. The seeds are small, black, and slightly triangular in shape [2–6]. In general, the morphology of the *A. paniculata* plant is characterized by its simple, unassuming appearance and its ability to adapt to a wide range of growing conditions.

The *A. paniculata* plant is native to South Asian countries, including India, Sri Lanka, and Pakistan. It is also found in other Southeast Asian countries, including Thailand, Indonesia, and Malaysia. The herb grows in a variety of soil types and climatic conditions, ranging from dry and arid to wet and humid environments. It is typically found in tropical and subtropical regions, where it is often cultivated as a medicinal crop. The plant is commonly found in the wild in areas with warm, humid climates, such as forest edges, riverbanks, and grasslands [7]. In recent years, the herb has gained popularity in other parts of the world, including Europe and North America, where it is often used as a dietary supplement to support the immune system and help prevent infections [8]. Although not widely cultivated outside its native range, cultivation practices are being developed to support its medicinal uses, with ongoing studies exploring its agricultural viability due to its relatively high yield.

Plant factories, typically with artificial lighting (PFAL), hydroponic or aeroponic growing systems, and precise climate control, create an optimal growing environment for plants. This allows for year-round crop production with higher yields and faster growth rates than traditional farming methods. The benefits of plant factories include reduced water usage, lower carbon emissions, and the ability to grow crops in urban areas where traditional agriculture is not feasible. Additionally, PFAL systems allow the production of crops free from pesticides, herbicides, and other contaminants. Plant factories are used to grow a variety of crops, including leafy greens, herbs, berries, and even some fruits and vegetables [9–12]. Closed systems are designed to increase production density, productivity, and resource utilization efficiency. High productivity is achieved by adjusting the interior environment to provide uniform lighting, temperature, and relative humidity while minimizing interaction with the outside environment. This also improves energy, water, and $CO_2$ efficiency [13]. Several studies have been conducted in closed production environments to determine optimal conditions for improved growth characteristics, higher yield, and nutritional quality of plants [14] reported that harvesting time and plant density significantly affect the quality of *A. paniculata*. Studies have shown that light affects physiological responses, antioxidant capacity, and chemical composition in holy basil [15].

Andrographolide, the principal bioactive compound in *A. paniculata*, is known for its diverse therapeutic properties, including anti-inflammatory, antiviral, antibacterial, and antioxidant effects [16,17]. This compound plays a crucial role in modulating the immune system, making it valuable in treating infections, inflammatory diseases, and certain cancers [18]. The lack of consistent andrographolide levels in crude drugs from various geographical sources impacts the quality of pharmaceutical manufacture and its therapeutic efficacy, posing a significant concern for the use of this

herb in medical therapies [19]. Further exploration into standardizing cultivation practices and genetic selection is essential to ensure uniform andrographolide content, which will enhance the reliability and effectiveness of *A. paniculata* -based pharmaceuticals.

The genetic variability of *A. paniculata* populations also plays a crucial role in andrographolide production. Several studies have investigated the genetic diversity of *A. paniculata* populations in different regions. The study found significant genetic diversity among the populations, indicating that the herb has a wide range of genetic variability within its species [20]. Another study in Thailand used molecular markers to analyze the genetic diversity of *A. paniculata* cultivars and found low genetic diversity, likely due to artificial selection and clonal propagation [21,22]. Further research is needed to clarify the plant's genetic makeup and its implications for medicinal applications and cultivation practices. Understanding genotype variations is crucial for developing improved cultivars and enhancing *A. paniculata* 'agricultural potential. One of the pivotal aspects of this study involves the use of Genome-Wide Association Studies (GWAS) to investigate the genetic bases of yield and bioactive content in *A. paniculata*. GWAS has been extensively utilized across various plant species to identify genetic loci associated with important traits, offering insights that are critical for breeding and genetic improvement. For example, Shariatipour et al. [23] conducted comparative genomic analyses in wheat that identified quantitative trait loci for micronutrient contents and grain quality, providing a similar framework for our analysis in *A. paniculata* [24]. Likewise, studies on other crops like rapeseed and cumin have utilized GWAS to uncover genetic elements linked to agronomic traits under stress conditions, further supporting the application of these methods to enhance medicinal plant yields and quality under varied environmental stresses [25–27].

The specific objectives of this study are to evaluate growth performance, crop productivity, and andrographolide content among local Thai *A. paniculata* accessions under PFAL conditions, investigate phylogenetic relationships using genome-wide SNP analysis, and conduct GWAS to identify genes associated with yield and bioactive compound production. These efforts aim to refine cultivation practices and genetic selection to standardize and enhance the pharmacological efficacy of *A. paniculata* -based treatments. By integrating findings from similar studies, this research contributes to a broader understanding of the genetic and environmental factors that influence medicinal plant productivity, setting a foundation for future innovations in the cultivation and application of *A. paniculata*.

## Materials and methods

### Plant materials and growth conditions

Seeds from six *A. paniculata* accessions, including Ratchaburi (RB), Phitsanulok (PL), Phichit (PC), Sa Kaeo (SK), Nakhon Pathom (NP), Songkhla (SH), and Prachin Buri (PB), were collected from seven provinces in Thailand. Additionally, commercially available CR (Farm Organic Seed, Wiang Papao, Chiang Rai Province), TTT (BENJAMITR ENTERPRIS (1991) CO., LTD., Bang Bua Thong, Nonthaburi Province), and TVRDC (Tropical Vegetable Research Center) as standard accession were also included (Table 1). This study did not require specific permits as the research was conducted entirely on private land with the owner's consent and in publicly accessible areas where no collection permits are mandated. All activities complied with local and national regulations governing plant research and collection.

The seed germination and growth conditions for *A. paniculata* followed a modified method from Chutimanukul et al. [14]. Seeds were planted on sponge trays containing reverse osmosis water at 100% moisture saturation. The trays were moved to shelves in a controlled environment with 100 µmol m$^{-2}$ s$^{-1}$ of white LEDs (AGRI-OPTECH Co., Ltd, Taiwan), a 16 h day$^{-1}$ photoperiod, 70 ± 5% relative humidity (RH), and 400 ± 50 µmol mol$^{-1}$ (ppm) $CO_2$ concentration for two weeks. Seedlings with fully expanded cotyledons were then transferred to a plant production room in the PFAL system under similar lighting and photoperiod conditions. The room was maintained at 70 ± 5% RH, with an airflow rate of 1.0 m s$^{-1}$, $CO_2$ concentration of 1000 ± 50 µmol mol$^{-1}$, and a temperature of 30 ± 3°C.

**Table 1. List of ten *A. paniculata* accessions from ten locations in Thailand with coding name of plants gown in PFAL system.**

| No. | Accession name | Location | Code |
|---|---|---|---|
| 1 | Ratchaburi | Ban Pong, Ratchaburi Province | RB |
| 2 | Phitsanulok | Phitsanulok Seed Research and Development Center, Wangthong, Phitsanulok Province | PL |
| 3 | Phichit | Agricultural Research and Development Center, Mueang, Phichit Province | PC |
| 4 | Sa kaeo | Mueang Sa Kaeo District, Sa kaeo Province | SK |
| 5 | Nakhon Pathom | Kamphaeng Saen District, Nakhon Pathom Province | NP |
| 6 | Songkhla | Muang District, Songkhla Province | SH |
| 7 | Prachin Buri | Chaophraya Abhaibhubejhr Hospital, Prachin Buri Province | PB |
| 8 | TVRDC | Tropical Vegetable Research Center of Kasetsart University, Kamphaeng Saen, Nakhon Pathom Province | TVRDC |
| 9 | Ching Rai | Farm Organic Seed, Wiang Papao, Chiang Rai Province | CR |
| 10 | Nonthaburi | BENJAMITR ENTERPRISE (1991) CO., LTD., Bang Bua Thong, Nonthaburi Province | TTT |

Seedlings were transplanted onto hydroponic foam boards at a density of 20 plants $m^{-2}$ and grown using the Deep Flow Water Technique (DFT). Environmental and growth conditions were monitored in real-time using a multi-sensor probe, and data were recorded every minute, with mean values calculated (S1 File). For the first two weeks, the electrical conductivity (EC) of the nutrient solution was set at 1.0 dS $m^{-1}$. Afterward, the EC was increased to 1.5 dS $m^{-1}$, and light intensity was raised to 300 µmol $m^{-2}$ $s^{-1}$ of PPFD. Plants were nourished with a modified Enshi solution (1:200), consisting of 190 g $L^{-1}$ $Ca(NO_3)_2$, 162 g $L^{-1}$ $KNO_3$, 98 g $L^{-1}$ $MgSO_4$, 30.8 g $L^{-1}$ $NH_4H_2PO_4$, 4 g $L^{-1}$ Fe-EDTA, 5 g $L^{-1}$ $H_3BO_3$, 0.572 g $L^{-1}$ $H_3BO_3$, 0.422 g $L^{-1}$ $MnSO_4 \cdot 4H_2O$, 0.044 g $L^{-1}$ $ZnSO_4 \cdot 7H_2O$, 0.016 g $L^{-1}$ $CuSO_4 \cdot 5H_2O$, and 0.005 g $L^{-1}$ $Na_2MoO_4 \cdot H_2O$ [14,28].

## Growth and biomass traits

To evaluate growth responses across the eight *A. paniculata* accessions, and plant characteristics were recorded at 30, 60, and 90 days after transplant (DAT). Plant height and width were measured using ImageJ software (ImageJ; http://imagej.nih.gov/ij/), while the number of leaves and stalks were manually counted. After harvesting, the above-ground tissue (stem, leaves, inflorescence) was weighed to obtain fresh weight (FW) using digital scales. Plants were then dried in an oven at 50°C for 72 hours, and the dry weight (DW) was measured.

## Sample extraction and Andrographolide analysis

At 30, 60, and 90 DAT, plant samples were freeze-dried using an SP VirTis Genesis Pilot Lyophilizer (SP Scientific, USA) for 48 hours. Extraction followed a modified method described by Chutimanukul et al. [14] and Pholphana et al. [29]. The freeze-dried samples were ground with a mortar and pestle to obtain a fine powder. For extraction, 120 mg of fine powder was mixed with 10 mL of methanol (99.9%, HPLC grade, Fisher), sonicated for 30 minutes at 25°C using an ultrasonic cleaner (Bransonic, Branson, Germany), and centrifuged at 5,000–7,000 rpm for 5 minutes (Benchtop centrifuge 5810 R, Eppendorf, USA). The supernatant was filtered through Whatman No. 1 filter paper, and the mixture was evaporated using a Genevac Rocket Centrifugal Evaporator (SP Scientific, USA) to obtain the crude extract.

The dried extract was dissolved in 10 mL by adding 5 mL of 5% methanol (HPLC grade, Fisher) and purified using a C18 solid-phase extraction Florisil 6 cc column (Waters, USA). For quantification of andrographolide (AP1), the supernatant was diluted 10-fold with 80% aqueous methanol, filtered using 0.22 µm syringe filters, and stored at -20°C for future analysis. The chemical composition of AP1 was determined using high-performance liquid chromatography (HPLC) (UltiMate™ 3000 UHPLC system, Thermo Scientific, USA) combined with a photodiode array detector (Dionex™ UltiMate™ 3000 Diode Array Detector, Thermo Scientific, USA), following the methods described by [14,29]. The separation was conducted on an ODS Hypersil C18 column (250 x 4.6 mm, 5 µm particle size, Thermo Scientific, USA) with acetonitrile HPLC gradient grade (Fisher) and deionized water from a Milli-Q water purification system, at a flow rate of 1.0 mL $min^{-1}$.

The injection volume was 10 µL, and detection occurred with a UV detector at 206 nm for 30 minutes. AP1 content was normalized by comparison with a standard calibration curve of andrographolide solution (Sigma-Aldrich), and the concentration of AP1 was expressed as milligrams per gram of DW.

## Statistical analysis

The experiments were conducted using a completely randomized design (CRD) with four replications, with each replication consisting of five plants per accession. To evaluate the data collected for each parameter, an analysis of variance (ANOVA) was performed. Mean comparisons across various growth parameters, biomass traits, and andrographolide content were analyzed using Duncan's Multiple Range Test (DMRT), with significance determined at $p \leq 0.05$.

To investigate the relationships among different phenotypic parameters, including plant height, bush width, number of stalks, number of leaves, and fresh and dry weights of above-ground tissue, principal component analysis (PCA) and hierarchical clustering analysis were performed using SAS software (SAS, Cary, NC, USA). A visual heat map was generated to facilitate easy inspection and interpretation of the resulting clusters.

## Sample collection and RADseq sequencing

The association panel consisted of a diverse collection of 62 Thai *A. paniculata* samples (Table 2) including four plants of CR, TTT and TVRDC, five plants of RB, seven plants of SK, NP and SH, eight plants of PL, PC and PB. Leaf tissues were collected with the national guidelines issued by the National Omics Center, NSTDA [30,31]. After sample collection, leaf tissues were frozen in liquid nitrogen and were stored at -80°C. Genomic DNA was extracted using the QIAGEN Genomic-tip 100/G by the manufacturer's protocol (Qiagen, Hilden, Germany). The amount of DNA was quantified using Qubit fluorometer (ThermoFisher Scientific, Waltham, USA), and the integrity of samples were assessed using a Pippin Pulse Electrophoresis System (Sage Science, Beverly, USA). One of *A. paniculata* accession was used as a reference genome sequence by the 10× Genomics technology with linked-read sequencing, which was based on microfluidics methods to construct long-read information from short-read sequencing (10× Genomics; https://www.10xgenomics.com). The genomics library was constructed from 1 ug of high-quality and high molecular weight of template DNA, following manufacturer's instructions on the Chromium Genome Library Kit and Gel Bead Kit v2, the Chromium Genome Chip Kit v2, and the Chromium i7 Multiplex Kit (10× Genomics). The sequencing library was performed using the Illumina HiSeq X Ten, and 150 bp of paired-end reads were generated.

For the restriction site associated DNA sequencing (RADseq) library, approximately 1 ug of DNA in each 61 *A. paniculata* accessions was used for a library construction following the MGIEasyTM RAD Library Prep Kit Instruction Manual (MGI Tech, Shenzhen, China). Restriction enzyme digestion (Taq1) was used to digest genomic DNA, and the DNA fragments obtained were ligated with a unique barcoded adapter. The samples were pooled in equimolar amount, then exposed to PCR and quantities were determined. Paired-end sequencing (150 bp) was carried out on the MGISEQ-2000RS following the manufacturer's protocol.

**Table 2. Shapiro tests for normality of phenotypic trait data. Fresh weight (FW) was log transformed attaining normality while dry weight (DW) was transformed using cube roots but did not attain a normal distribution. p < 0.05 indicates a violation of normally distributed data.**

| Trait | Statistic | p-value | Transformation | Statistic (transformed) | p-value (transformed) |
|-------|-----------|---------|----------------|-------------------------|-----------------------|
| AP1 | 0.982 | .490 | – | – | – |
| FW | 0.933 | **0.002**\*\* | log | .970 | 0.136 |
| DW | 0.930 | **0.002**\*\* | cube root | .958 | **0.034**\* |

## Variant calling

Read quality was evaluated using FastQC 1.0.0 [32] with the default parameters. Sequenced reads were aligned to the reference genome [33] using Bowtie2 [34]with default parameters. Subsequent processing, including duplicate removal was performed using Samtools [35] and the PICARD command line suite of tools (http://picard.sourceforge.net). Finally, variant calls to yield the final matrix were filtered using VCFtools 0.1.16 [36] with the following parameters: minimum minor allele frequency (MAF) > 0.01, max missing < 0.5, and Hardy-Weinberg significance set at 0.05.

## Genetic diversity evaluation

To investigate the genetic relationships of *A. paniculata* accessions, principal component analyses (PCA) were conducted using the *sklearn* Python package [37] while we then used the Python *pyclustering* library (www.pypi.org) to evaluate the most likely number of distinct genetic clusters. Maximum likelihood (ML) and neighbour joining phylogenetic reconstruction were performed using iQtree software [38] and scripts from the riceExplorer pipeline resource [39], respectively. iQtree was employed to construct ML trees with 1,000 bootstraps. "-m MFP+ASC" was set for automatic searching the optimal model of nucleotide substitution. The bootstrap value was adjusted by "-bnni". The vcf matrix was subsequently thinned to exclude calls less than 10000 bp apart. We then conducted linkage disequilibrium decay analyses to evaluate chromosomal signatures of recombination patterns. Finally, we calculated genetic diversity indices (F-statistics) using the Stacks Populations utility [40]. Evaluated indices: observed and expected homozygosity and heterozygosity, nucleotide diversity ($\pi$), and $F_{IS}$ and pairwise $F_{ST}$.

## Mantel test and genome-wide association (GWAS) analysis

To test general relationships between accession genotype and phenotypic trait assays we conducted mantel tests using the '*mantel*' library (https://github.com/jwcarr/mantel) in Python using default settings. We evaluated pairwise genetic distances by Kimura-2-parameter (K2P) model [41] between samples (calculated in riceExplorer)[39] against distance matrices computed from each individual trait assay and for all traits combined.

We used the GWASpoly library in R (https://github.com/jendelman/GWASpoly) to conduct GWAS analyses examining both the 'additive' and 'general' models with default settings. We passed output files to custom Python scripts to generate QQ-plots by plotting observed probabilities for each marker against the set of probabilities at which to evaluate the inverse distribution. We further generated Manhattan plots to evaluate individual marker associations by plotting each marker against the negative logarithm of its GWASpoly-generated probability. For each plot, we calculated the false discovery rate at 5% threshold [42] and the Bonferroni correction threshold is also at 5% [43]

# Result

## Phenotypic traits evaluation

We evaluated four growth parameters, stem height, plant width, stalk number, and leaf number along with fresh weight (FW), dry weight (DW), and andrographolide content (AP1) across ten locals *A. paniculata* accessions at vegetative (30 DAT), initial flowering (60 DAT) and harvesting (90 DAT) stages. Significant differences in stem height were observed among the accessions at different development stages (Fig 1A). PB consistently showed higher plant height compared to other accessions at all stages of plant development. Specifically, PB and TTT had the highest plant heights at 90 DAT, measuring 78.38 ± 2.68 cm and 79.69 ± 2.76 cm, respectively, while CR and SH had the lowest mean heights. For plant width, SK had a significantly greater value (40.81 ± 3.20 cm) compared to other accessions at 60 DAT (Fig 1B). At 90 DAT, TTT exhibited the highest plant width (53.00 ± 0.89 cm). The greatest stalk number was obtained when *A. paniculata* was developed at 90 DAT. The greatest stalk number was recorded in PC, SK and SH while the lowest stalk number was observed in RB and the commercial TTT accession (Fig 1C). Among ten accessions, NP and PB displayed the highest

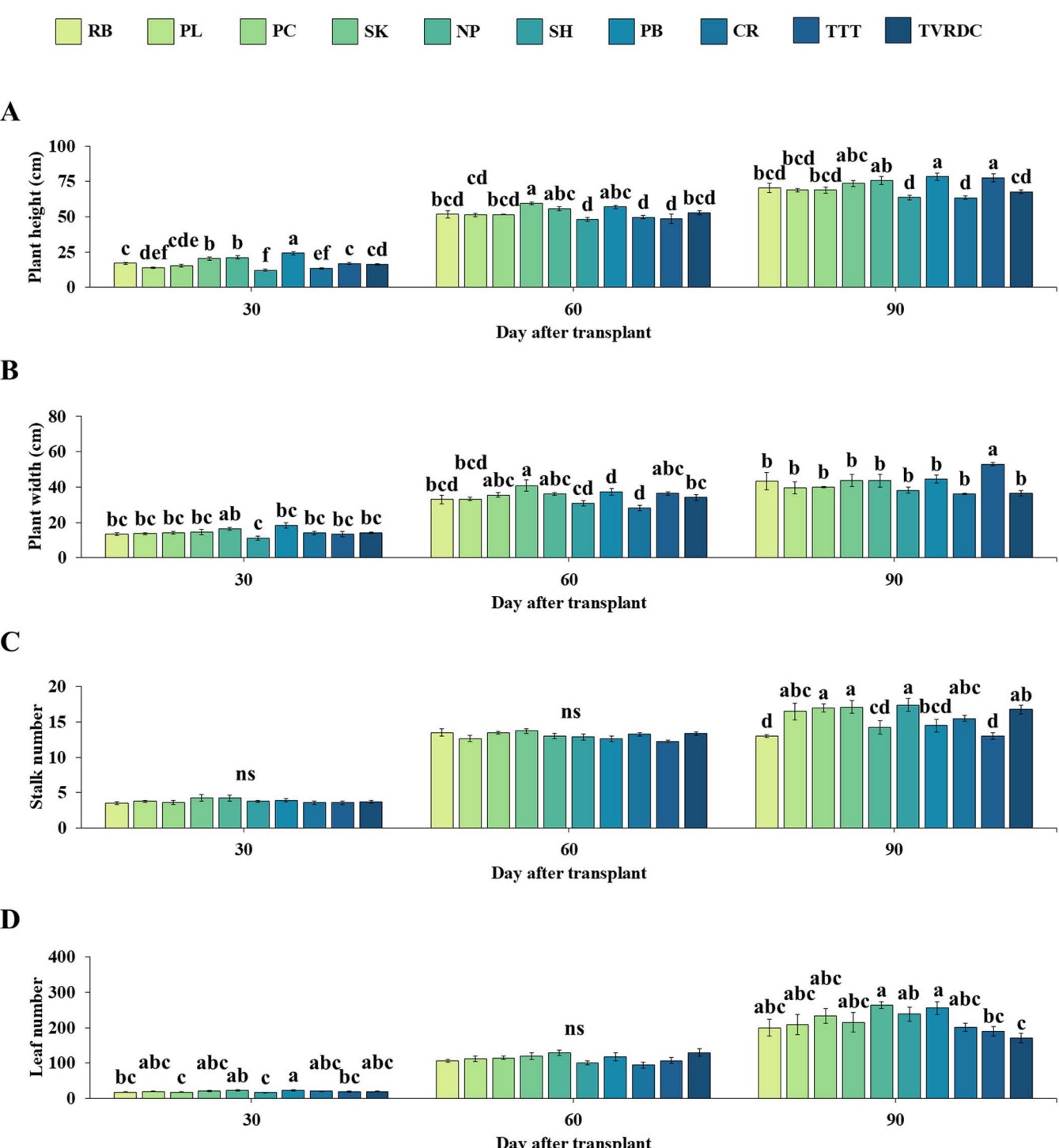

**Fig 1. Growth characteristics of *A. paniculata* across three stages.** Plant height (A), plant width (B), number of stalks (C) and number of leaves (D) of 10 local *A. paniculata* accessions at vegetative (30 DAT), initial flowering (60 DAT) and harvesting (90 DAT) stages. Values are represented as mean ± SE (n = 4). Bars represent standard error. ANOVA was performed, followed by a mean comparison with DMRT. Different letters indicate significant difference between accessions at p < 0.05. "ns" indicates no significant difference.

leaf number per plant than two commercial accessions while TTT and TVRDC had a significantly lower leaf number (Fig 1D) at 90 DAT.

At the harvesting stage, significant differences in FW of above-ground tissue were observed among accessions (Fig 2A). TTT displayed the highest FW, with a mean of 142.38 ± 9.99 g per plant. DW followed a similar trend, with TTT showing the highest value at 25.79 ± 1.76 g per plant (Fig 2B). The morphological traits of the ten accessions at the harvesting stage are illustrated in Fig 3.

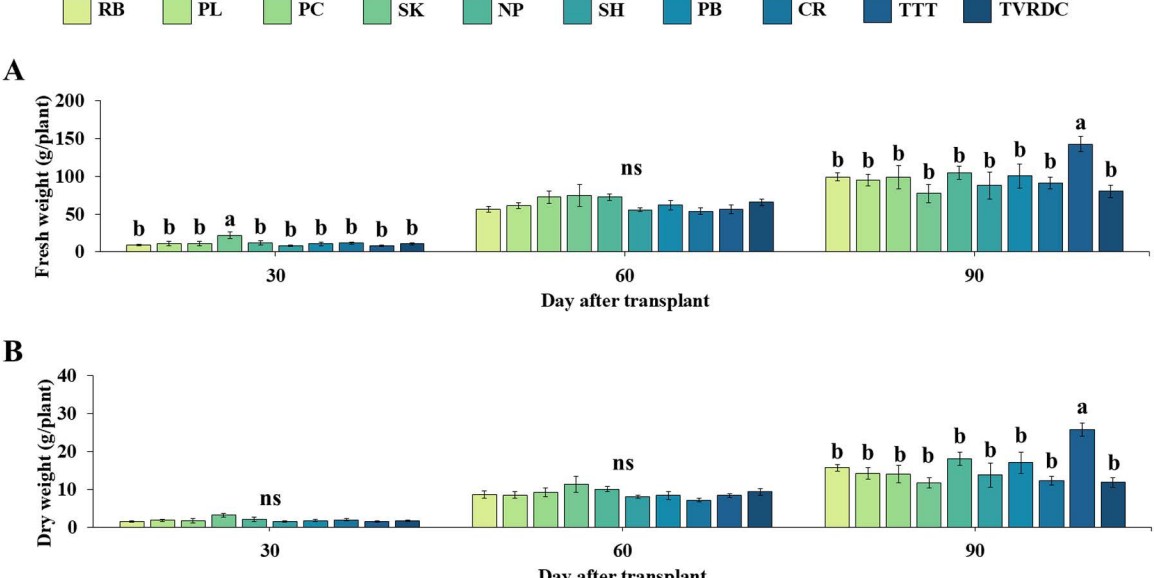

**Fig 2. Biomass acuumulation of *A. paniculata* across three stages.** Fresh weight (A) and dry weight (B) of above-ground tissue of 10 local *A. paniculata* accessions at vegetative (30 DAT), initial flowering (60 DAT) and harvesting (90 DAT) stages. Values are represented as mean ± SE (n = 4). Bars represent standard error. ANOVA was performed, followed by a mean comparison with DMRT. Different letters indicate significant difference between accessions at p < 0.05. "ns" indicates no significant difference.

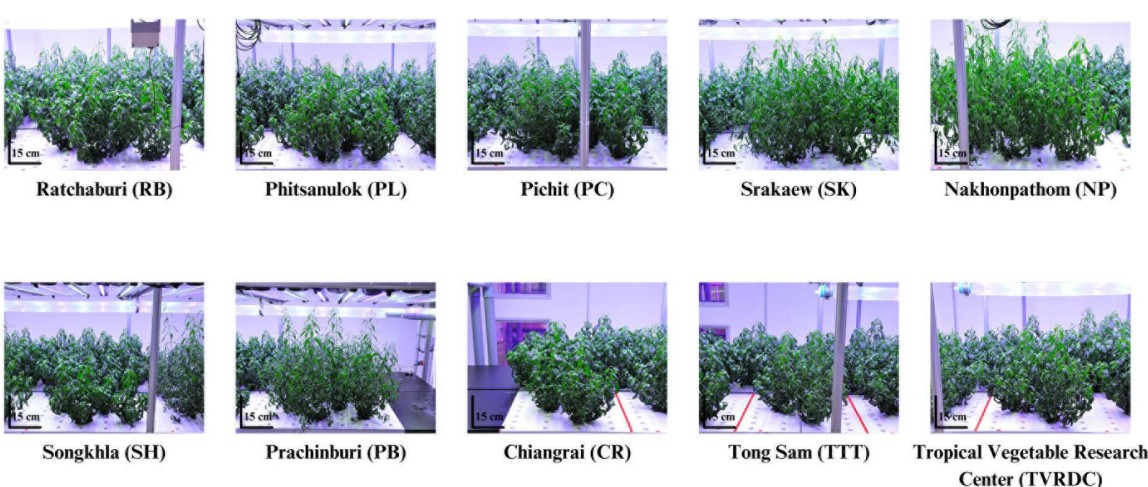

**Fig 3. The morphology of 10 local *A. paniculata* accessions at harvesting (90 DAT) stages.**

Additionally, andrographolide content (AP1) was assessed in all ten accessions at each developmental stage using GC analysis (Fig 4). Variation among 10 accessions showed a significant difference level of AP1 at 60 DAT. The amount of AP1 found in all the cultivars/accessions ranged from 21.11 to 32.27 mg g$^{-1}$ DW. The above-ground tissue of RB had the highest accumulation of AP1 (32.27 mg g$^{-1}$ DW). While the level of AP1 was not significantly different among 10 accessions at 30 and 90 DAT.

## Hierarchical clustering analysis of the phenotypic traits

The growth characteristics of ten local *A. paniculata* accessions, including RB, PL, PC, SK, NP, SH, PB, and TVRDC, along with the commercial control varieties CR and TTT, were evaluated in a plant factory with artificial lighting (PFAL) under controlled environmental conditions at the vegetative (30 DAT), initial flowering (60 DAT), and harvesting (90 DAT) stages.

Heat mapping and hierarchical clustering analyses of six growth parameters and andrographolide content (AP1) for these ten accessions across developmental stages are presented in Fig 5. The analyses grouped parameters according to their respective developmental stages. Cluster 1 represents the vegetative stage and is further divided into two sub-clusters: the first sub-cluster includes SH, TTT, RB, and SK, while the second sub-cluster comprises PL, TVRDC, CR, PC, NP, and PB. Cluster 2 corresponds to the initial flowering stage, which also splits into two sub-clusters. The first sub-cluster consists of CR, SH, TTT, PL, PC, SK, and RB, while the second sub-cluster groups TVRDC, NP, and PB. Interestingly, Cluster 3, representing the harvesting stage, is likewise divided into two sub-clusters. The first sub-cluster includes TVRDC, CR, SK, SH, PL, PC, RB, PB, and NP, while the second sub-cluster consists solely of TTT. This clustering pattern highlights the distinct growth and AP1 accumulation characteristics across the developmental stages, with certain accessions clustering more closely together, particularly at the vegetative and harvesting stages.

## Agronomic traits assay

The data distributions of three phenotypic traits, measured for subsequent GWAS analyses, were assessed for statistical normality, and transformations were applied where necessary to optimize normality. Fresh weight (FW) and dry weight (DW) required transformation (Table 2). Fig 6 presents boxplot distributions of the transformed data.

## Genetic diversity evaluation and phylogenetic analysis

Sequencing identified 109,847 variants across the 62 *A. paniculata* samples. After filtering this was reduced to 16,431 variant calls, of which 5,829 were SNPs and 10,602 were indels. Maximum likelihood and neighbour-joining phylogenetic reconstruction of variant calls both indicate two well-supported major clades among 62 *A. paniculata* samples in Thailand (Fig 7). In general, phylogeny indicates non-extreme levels of geographic clustering, but a particular well-defined clade

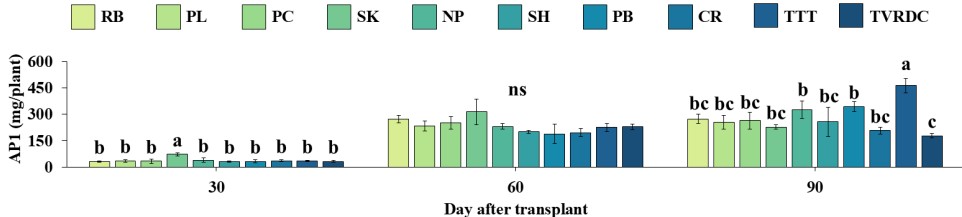

**Fig 4. Andrographolide (AP1) content of *A. paniculata* across three stages.** AP1 content (mg DW$^{-1}$ m$^{-2}$) of *A. paniculata* above-ground tissue of 10 local *A. paniculata* accessions at vegetative (30 DAT), initial flowering (60 DAT) and harvesting (90 DAT) stages. Values are represented as mean ± SE (n = 4). Bars represent standard error. ANOVA was performed, followed by a mean comparison with DMRT. Different letters indicate significant difference between accessions at p < 0.05. "ns" indicates no significant difference.

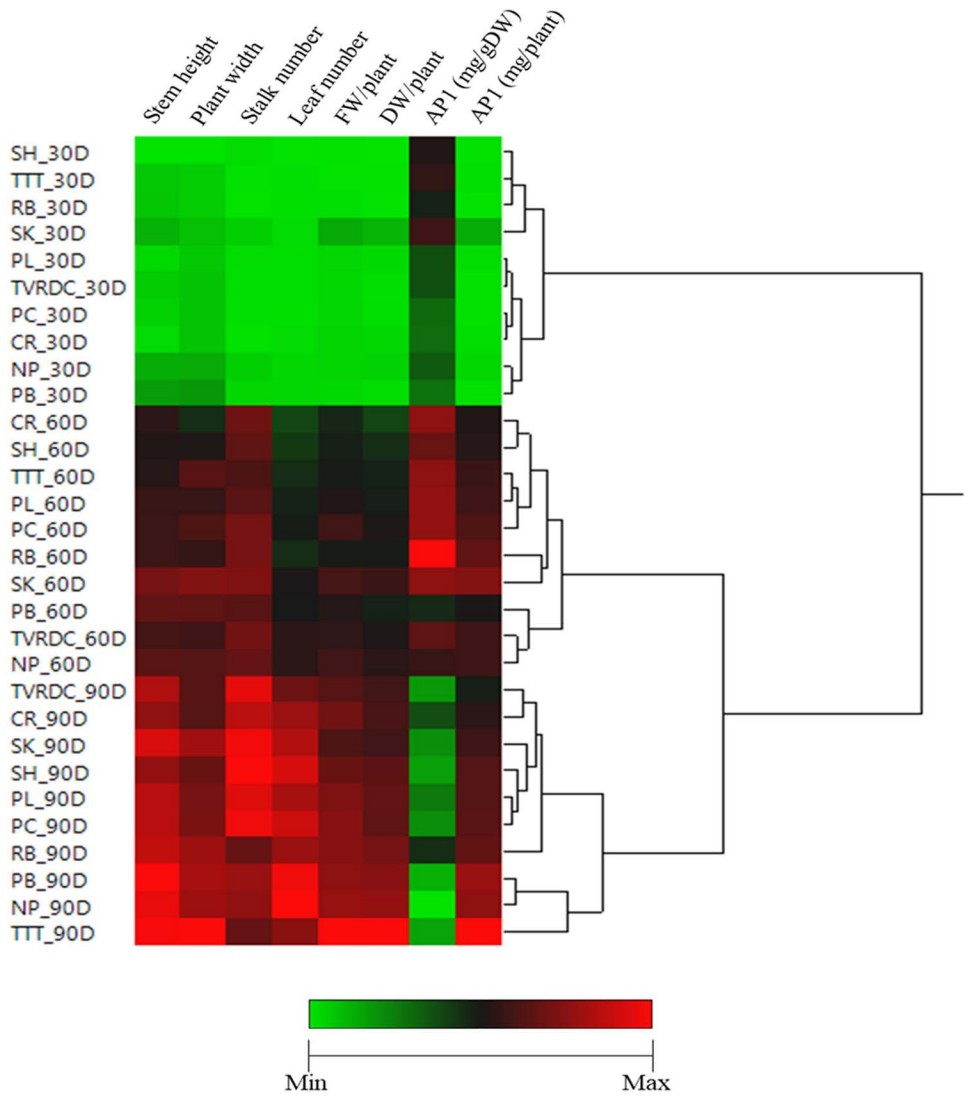

**Fig 5. Heat mapping and clustering analysis of all measured variables using Z-scores for the normalized value.**

includes southern and lower northern samples, possibly indicating transplantation of *A. paniculata* stock. This is also borne out by PCA analyses (Fig 8) where Xmeans clustering evaluation indicates that genomic structure comprises two demes that are loosely correlated with geography including a grouping of southern and lower northern samples.

## Population genetic (F-statistics) results

Population genomic analyses (including observed and expected homozygosity and heterozygosity, nucleotide diversity, and $F_{IS}$) indicate low levels of standing genomic diversity at the population level (Table 3). For example, nucleotide diversity ($\pi$) mostly ranges from 0.050–0.064. Accordingly, observed homozygosity and inbreeding coefficients $F_{IS}$ are low. Pairwise (between populations) $F_{ST}$ is also low (typically ca. ≤0.1; Table 4) among natural populations from plants sampled in Thailand, indicating homogenized populations. However, $F_{ST}$ between Southern and Western populations reach moderately high values of 0.153 indicating some separation.

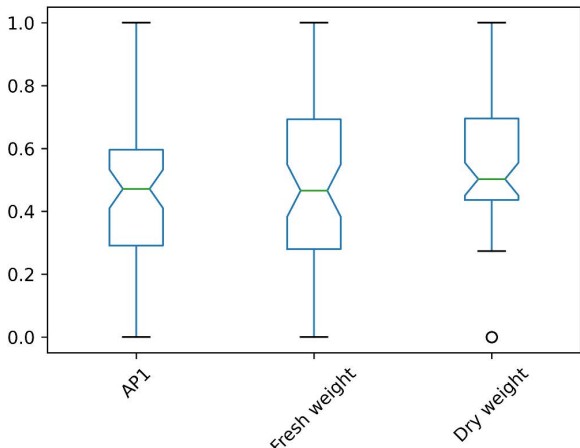

**Fig 6. Boxplots of standardized (*z-score*) data distributions.** Fresh and dry weights are transformed. Despite the transformation, the dry weight is not normally distributed.

### Mantel test and genome-wide association analyses (GWAS) evaluation

Mantel tests of matrices for Kimura-2-parameter (Kimura 1980) pairwise genetic distances against distance matrices for all individual and combined phenotypic measurements indicate that *A. paniculata* accession genotypes predict fresh and dry weights (Table 5), as well as the combined phenotypes.

GWAS analyses indicated only two variants (both SNP and indel) displaying statistical associations with trait data were observed for AP1 synthesis (Figs 9 and 10). These significantly segregating variants are found on chromosome 9 and an unidentified scaffold from the reference genome.

## Discussion

This study highlights significant phenotypic diversity among local *A. paniculata* accessions grown under controlled environmental conditions. The evaluation of key growth traits, including stem height, plant width, stalk number, and leaf number, across three developmental stages (vegetative, initial flowering, and harvesting) demonstrates the influence of genetic variability on plant development. Our results align with previous studies that also reported substantial morphological differences among *A. paniculata* accessions under varied environmental conditions [44]. Building on our findings, the notable height and robustness of specific accessions such as PB and TTT suggest their potential for high biomass yield, essential for medicinal applications. This observation is in harmony with [24] and Shariatipour et al. [27], who similarly highlighted the correlation between genetic traits and biomass productivity in medicinal plants as well as in wheat, demonstrating how such phenotypic traits can directly influence the economic viability of plant species by enhancing the volume of extractable medicinal compounds [23,45]. Our study contributes to this discussion by confirming that morphological traits like plant height, width, and stalk number not only signify a plant's adaptive responses but also its commercial potential.

In terms of plant architecture, traits such as plant width and stalk number exhibited notable variation. For instance, SK displayed the greatest plant width at the initial flowering stage (60 DAT), while PC, SK, and SH recorded the highest stalk numbers at the harvesting stage (90 DAT). This variation in stalk number is likely correlated with the plant's structural capacity to support increased biomass and possibly higher levels of medicinal compounds. These results are in line with findings by Chutimanukul et al. [14], who identified a positive correlation between stalk number and biomass accumulation in *A. paniculata*. Such structural traits are essential for plants intended for high-yield cultivation, as they enhance the plant's overall resilience and support during growth.

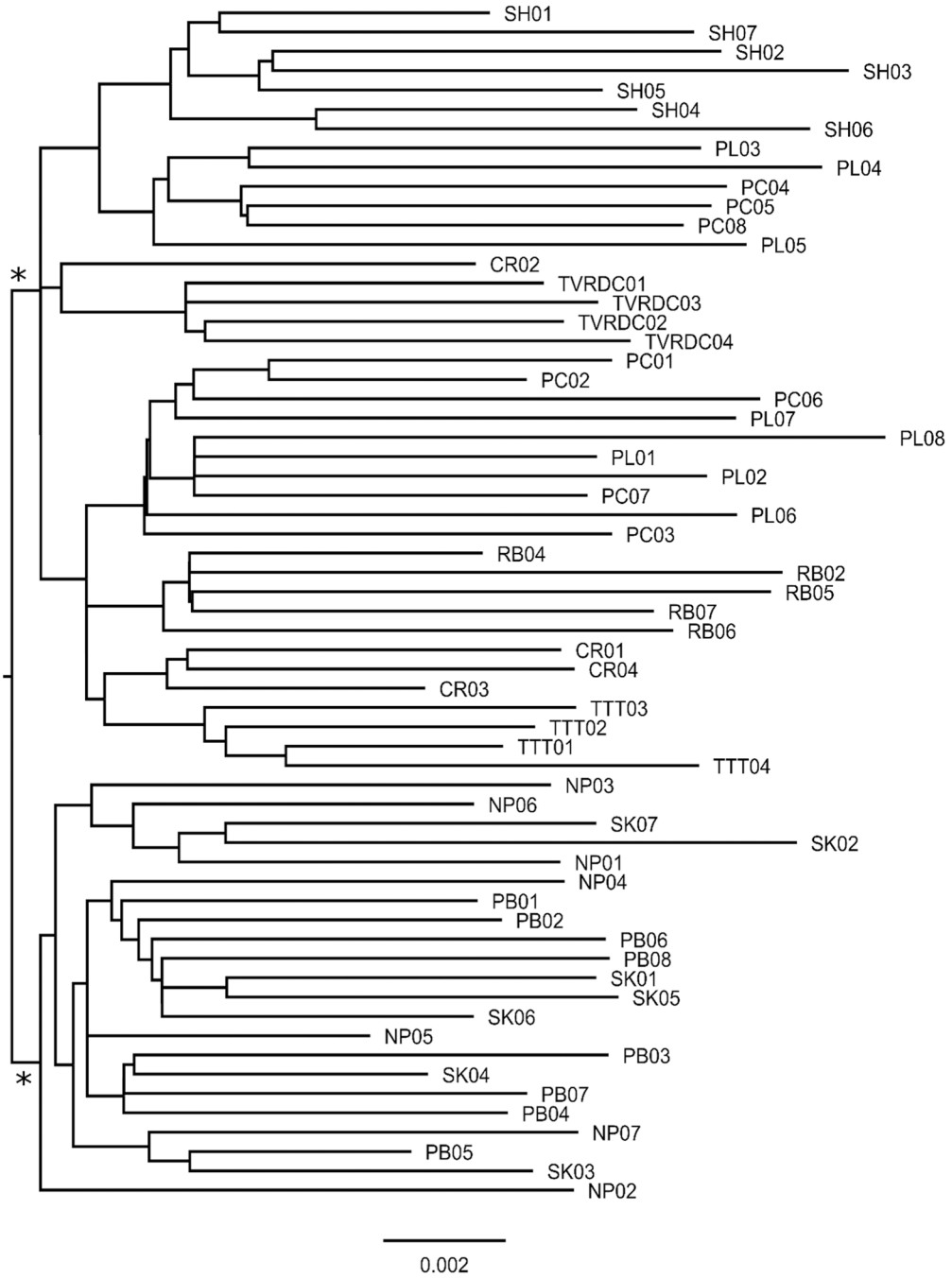

**Fig 7. Phylogenetic tree of _A. paniculata_ samples based on SNP markers.** Maximum likelihood phylogenetic reconstruction of 5,831 SNP markers across 62 _A. paniculata_ samples in Thailand. Two main clades (with 95% support for monophyly) are indicated.

The fresh and dry weight measurements further support the commercial potential of certain accessions. The high fresh and dry weights of TTT at the harvesting stage suggest that this accession may be particularly well-suited for large-scale production due to its biomass efficiency. This aligns with Akbar [46], who emphasized that biomass yield is a key

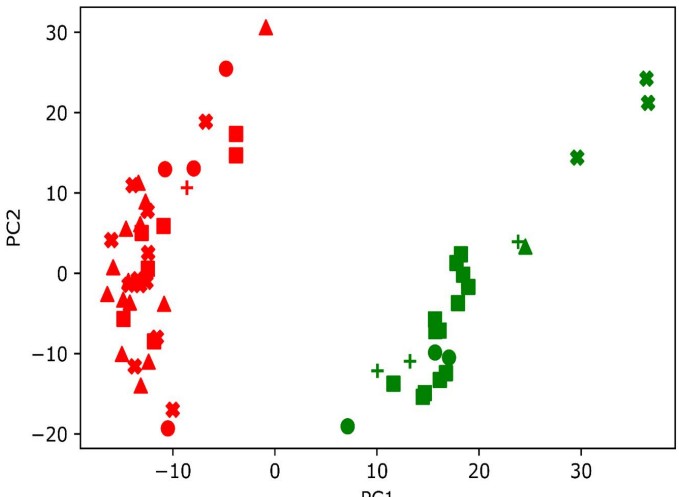

**Fig 8. Principal component analyses of SNP and Indel variants. a) for 62 *A. paniculata* samples in Thailand.**

**Table 3. Within population homozygosities, heterozygosities, nucleotide diversity (π) and inbreeding coefficients ($F_{IS}$).**

| Pop | n | Obs. Het | Obs. Hom | Exp. Het | Exp. Hom | Pi | $F_{is}$ |
|---|---|---|---|---|---|---|---|
| North | 20 | 0.014±0.0 | 0.986±0.0 | 0.062±0.01 | 0.938±0.01 | 0.064±0.01 | 0.358±0.23 |
| Central | 15 | 0.013±0.01 | 0.987±0.01 | 0.053±0.01 | 0.947±0.01 | 0.055±0.01 | 0.223±0.18 |
| East | 15 | 0.015±0.01 | 0.985±0.01 | 0.048±0.01 | 0.952±0.01 | 0.05±0.01 | 0.197±0.16 |
| West | 5 | 0.011±0.01 | 0.989±0.01 | 0.046±0.02 | 0.954±0.02 | 0.053±0.02 | 0.101±0.09 |
| South | 7 | 0.016±0.01 | 0.984±0.01 | 0.046±0.01 | 0.954±0.01 | 0.051±0.02 | 0.097±0.09 |

**Table 4. Pairwise population $F_{ST}$ relationships.**

| | Central | East | West | South |
|---|---|---|---|---|
| **North** | 0.037 | 0.048 | 0.053 | 0.054 |
| **Central** | | 0.051 | 0.078 | 0.078 |
| **East** | | | 0.103 | 0.095 |
| **West** | | | | 0.153 |

**Table 5. Mantel tests between phenotypic and genotypic distance matrices. Results for fresh and dry weights, and combined phenotypes are significant.**

| Trait | Correlation (%) | p-value | z-score |
|---|---|---|---|
| AP1 | -0.009 | 0.549 | -0.111 |
| FW | 0.153 | **0.015*** | 2.162 |
| DW | 0.175 | **0.012*** | 2.229 |
| all | 0.138 | **0.033*** | 1.813 |

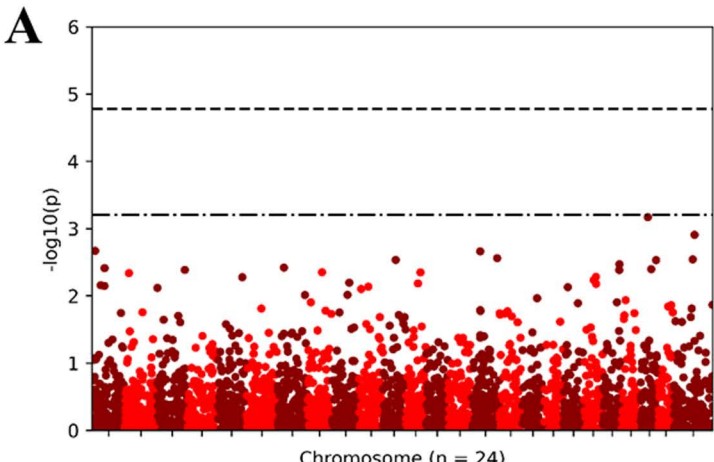

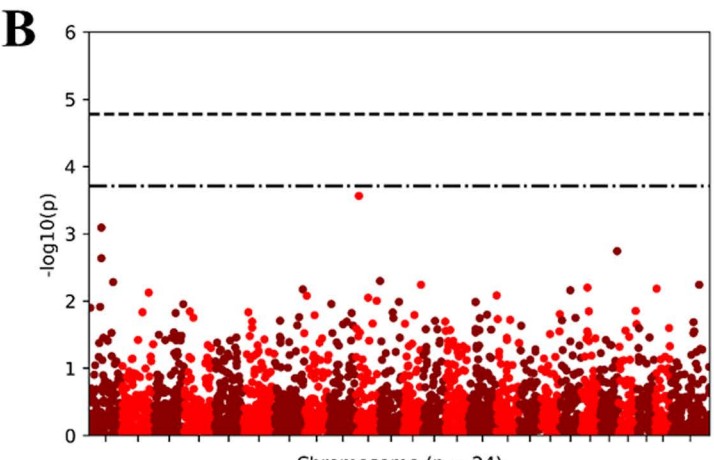

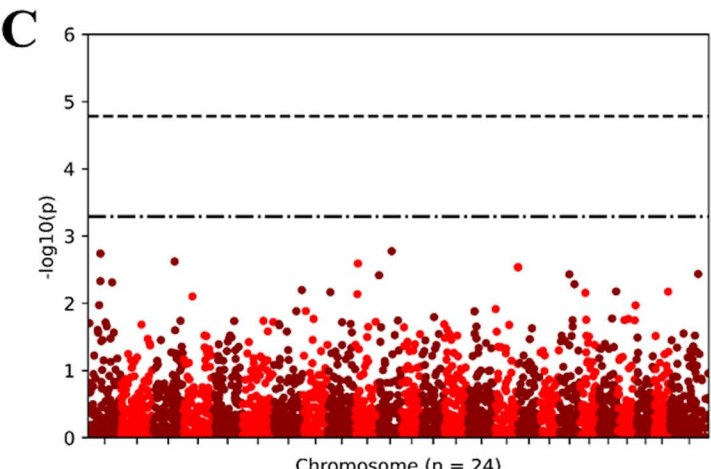

**Fig 9. GWASpoly performed GWAS analyses (Additive models) for six phenotypic trait assays.** Andrographolide (AP1) content (A), fresh weight (B), and dry weight (C).

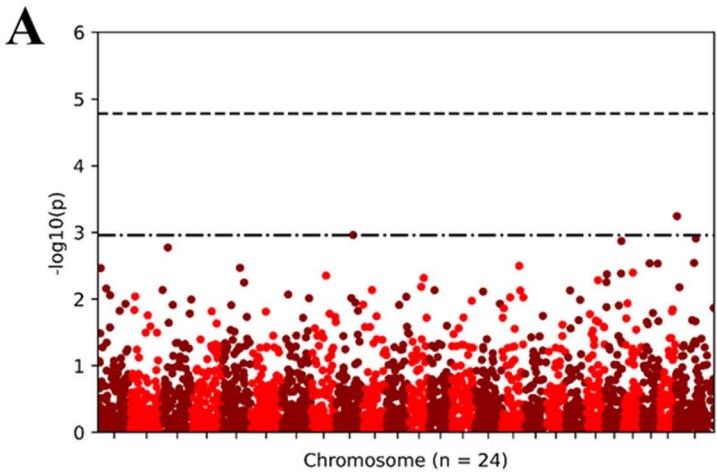

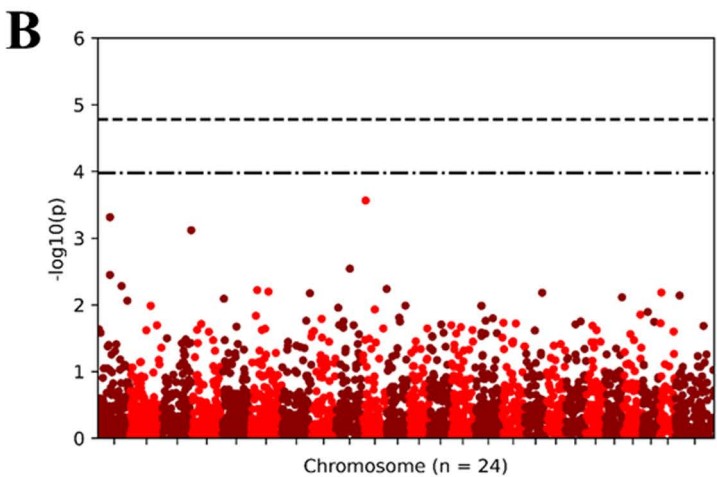

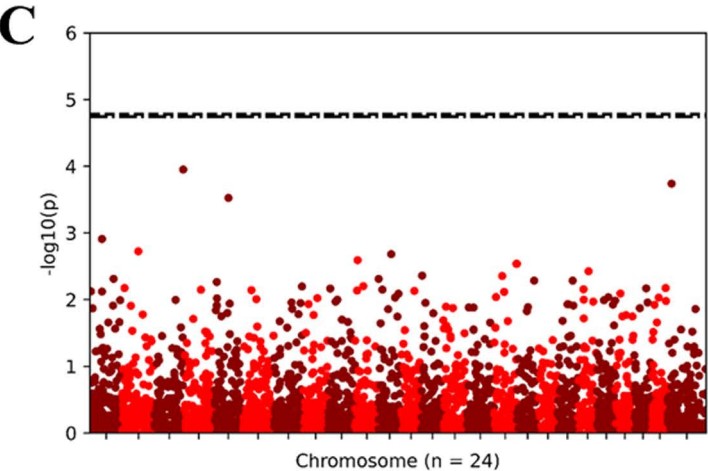

**Fig 10. GWASpoly performed GWAS analyses (General models) for six phenotypic trait assays.** Andrographolide (AP1) content (A), fresh weight (B), and dry weight (C).

determinant of economic value in *A. paniculata*, especially for medicinal applications where biomass volume directly impacts the quantity of extractable compounds.

For andrographolide content (AP1), significant variation was observed among the accessions, with RB showing the highest AP1 accumulation at 60 DAT. This indicates that the RB accession could be particularly valuable for medicinal use during the flowering stage, where AP1 levels peak. Our analysis of andrographolide content reveals significant inter-accessional variability, particularly highlighting the RB accession as a standout during the flowering stage. This aligns with the findings of Chutimanukul et al. [14], who noted genetic influences on compound synthesis in *A. paniculata*. Further supported by Li et al. [47], these insights underscore the critical impact of selecting the right developmental stage for harvesting to maximize the yield of active compounds. The ability to pinpoint such optimal harvesting times offers a strategic advantage for pharmacological uses, ensuring the maximum efficacy of the extracted compounds. Salami et al. [24], Salami et al. [25], Salami et al. [48] reinforce this point by illustrating the integration of GWAS, metabolomics, and transcriptomics in identifying key genetic markers linked to phenolic acid and flavonoid production in rapeseed under stress conditions.

Hierarchical clustering and heatmap analyses provided further insights into the developmental and phenotypic relationships among the accessions. The clustering patterns, particularly those observed at the vegetative and harvesting stages, reveal distinct growth trajectories among the accessions. For example, at the vegetative stage, two main subgroups emerged, with accessions such as SH, TTT, and RB clustering closely together, suggesting shared genetic or physiological adaptations to the controlled environmental conditions. Rahimi et al. [49] employed similar techniques to demonstrate how environmental and controlled growth conditions influence plant phenotypes and genetic expression. Our results corroborate these findings and extend them by identifying genetic markers through GWAS, as explored by Chen et al. [50], which could serve as potential targets for genetic enhancement programs focused on increasing the production of bioactive compounds. Additionally, Archangi et al. [26], Archangi et al. [51] provide context on how assessing genetic diversity and genotype selection in crops like cumin under drought stress can inform similar strategies in medicinal plants.

Genetic diversity analyses revealed relatively low levels of genetic differentiation among populations, particularly between Southern and Western populations. The observed low levels of genetic differentiation among populations suggest potential challenges in maintaining genetic diversity, a concern also raised by Chaturvedi et al. [44]. Addressing these challenges will require innovative breeding strategies that not only introduce new genetic material but also capitalize on the existing genetic variations to enhance both resilience and productivity. By recognizing these patterns and integrating them into breeding programs, researchers can more effectively tailor cultivation practices to produce phenotypically diverse and genetically robust plants, thereby ensuring sustainable production and higher pharmacological value.

Finally, the genome-wide association study (GWAS) identified two significant genetic variants associated with andrographolide content, providing potential markers for breeding programs aimed at enhancing the production of medicinal compounds. These SNP and indel variants located on chromosome 9 represent an important advancement in understanding the genetic underpinnings of andrographolide biosynthesis in *A. paniculata*. This discovery aligns with recent research in medicinal plants, where GWAS has proven effective in identifying genetic loci responsible for bioactive compound production [50]. Our findings highlight the complex interplay between genetics, phenotype, and environmental factors in *A. paniculata*, offering valuable perspectives for agricultural and pharmacological applications. As the demand for medicinal plants continues to grow, the insights from this study could inform more targeted breeding strategies, focusing on both enhancing genetic diversity and optimizing medicinal compound production. Future research should explore the potential for cross-regional genetic studies to further diversify genetic pools and investigate the molecular mechanisms underlying the biosynthesis of key medicinal compounds, ultimately leading to more effective and sustainable agricultural practices.

## Conclusions

This study assessed ten *A. paniculata* cultivars, including eight Thai and two commercial varieties (TTT and red CR), under Plant Factory with Artificial Lighting (PFAL) systems to determine their potential for high yield and bioactive

compound production. Notably, the TTT cultivar demonstrated superior morphological traits and high biomass at the 90-day harvest, making it particularly valuable for pharmaceutical applications. Phylogenetic analysis using 5,831 SNP markers highlighted that TTT and other Thai accessions such as CR, RB, PL, and PC share a close genetic relationship and exhibit potential for enhanced productivity in controlled environments. A genome-wide association study (GWAS) pinpointed a significant region on chromosome 9 crucial for andrographolide biosynthesis, suggesting a targeted focus for future breeding programs. These findings offer strategic insights for optimizing *A. paniculata* cultivation, potentially improving the efficiency of pharmaceutical crop production.

## Supporting information

**S1 File.  The details of the daily environment and growth conditions under PFAL system.**
(XLSX)

## Acknowledgments

The authors thank the valuable contributions for data collection provided by Miss Akira Thongtip. We are also grateful to Dr. Theerayut Toojinda, senior researcher at BIOTEC, and Prof. Dr. Poonpipope Kasemsap from the Horticulture Innovation Lab Regional Center Kasetsart University, for their invaluable conceptual guidance during the project planning phase.

## Author contributions

**Conceptualization:** Panita Chutimanukul.

**Data curation:** Praderm Wanichananan, Supattana Janta, Tanawut Chiangklang, Kriengkrai Mosaleeyanon.

**Formal analysis:** Praderm Wanichananan, Supattana Janta, Suchalee Sueachuen, Tanawut Chiangklang, Kriengkrai Mosaleeyanon, Siripar Korinsak, Clive Terence Darwell.

**Funding acquisition:** Praderm Wanichananan.

**Investigation:** Praderm Wanichananan, Supattana Janta, Suchalee Sueachuen, Tanawut Chiangklang, Kriengkrai Mosaleeyanon, Siripar Korinsak, Clive Terence Darwell, Panita Chutimanukul.

**Methodology:** Praderm Wanichananan, Supattana Janta, Panita Chutimanukul.

**Resources:** Praderm Wanichananan.

**Supervision:** Panita Chutimanukul.

**Validation:** Praderm Wanichananan, Siripar Korinsak, Panita Chutimanukul.

**Visualization:** Praderm Wanichananan, Clive Terence Darwell, Panita Chutimanukul.

**Writing – original draft:** Panita Chutimanukul.

**Writing – review & editing:** Panita Chutimanukul.

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
