## [Decision Letter · Decision Letter 0]

4 Feb 2025

PONE-D-24-60023Morphological characteristics and genome-wide association analysis among local Andrographis paniculata from Thailand under controlled environment in plant factoryPLOS ONE

Dear Dr. Chutimanukul,

Thank you for submitting your manuscript to PLOS ONE. After careful consideration, we feel that it has merit but does not fully meet PLOS ONE’s publication criteria as it currently stands. Therefore, we invite you to submit a revised version of the manuscript that addresses the points raised during the review process.

**ACADEMIC EDITOR: Please insert comments here and delete this placeholder text when finished.** The reviewers suggested several minor and major comments. I suggest major revisions. Kindly check the journal's technical requirements and formatting according to journal requirements.. 

We look forward to receiving your revised manuscript.

Kind regards,

Faham Khamesipour, Ph.D.

Academic Editor

PLOS ONE

2. Thank you for stating the following financial disclosure:  [This research was funded by the support of National Science and Technology Development Agency, Thailand (P2351505) and Thailand Basic Research Fund: fiscal year 2023 with Contract no. 4709540.].  Please state what role the funders took in the study.  If the funders had no role, please state: "The funders had no role in study design, data collection and analysis, decision to publish, or preparation of the manuscript." If this statement is not correct you must amend it as needed. Please include this amended Role of Funder statement in your cover letter; we will change the online submission form on your behalf.

Additional Editor Comments:

The reviewers suggested several minor and major comments. I suggest major revisions. Kindly check the journal's technical requirements and formatting according to journal requirements.

Reviewers' comments:

Reviewer's Responses to Questions

**Comments to the Author**

1. Is the manuscript technically sound, and do the data support the conclusions?

Reviewer #1: Yes

Reviewer #2: Yes

2. Has the statistical analysis been performed appropriately and rigorously? 

Reviewer #1: Yes

Reviewer #2: Yes

3. Have the authors made all data underlying the findings in their manuscript fully available?

Reviewer #1: Yes

Reviewer #2: Yes

4. Is the manuscript presented in an intelligible fashion and written in standard English?

Reviewer #1: Yes

Reviewer #2: Yes

5. Review Comments to the Author

Reviewer #1: 1-As one of important parts of the paper is about GWAS analysis, it is suggested to address high quality publications about the merits of GWAS analysis in different species in including medicinal plants. Bellow, several publications about GWAs and meta-analysis can be addressed in introduction as literature review and in discussion part for comparing similar results between the publications and the submitted manuscript.

Shariatipur et al. 2021. Comparative Genomic Analysis of Quantitative Trait Loci Associated With Micronutrient Contents, Grain Quality, and Agronomic Traits in Wheat (Triticum aestivum L.). frontiers in Plant Science, //doi.org/10.3389/fpls.2021.709817

Shariatipur et al 2021. Meta-analysis of QTLome for grain zinc and iron contents in wheat (Triticum aestivum L.). Eyphytica 217, //doi.org/10.1007/s10681-021-02818-8

Shariatipour et al. 2021. Genomic analysis of ionome-related QTLs in Arabidopsis thaliana. Scientific Reports, 11. doi.org/10.1038/s41598-021-98592-7

Salami et al. 2022. Comparative profiling of polyphenols and antioxidants and analysis of antiglycation activities in rapeseed (Brassica napus L.) under different moisture regimes. Food Chemistry, 399: //doi.org/10.1016/j.foodchem.2022.133946

Salami et al. 2023. Integration of genome wide association studies (GWAS), metabolomics and transcriptomics reveals phenolic acids and flavonoids associated genes and their regulatory elements under drought stress in rapeseed flowers. Frontiers in Plant Science, 14, 10.3389/fpls.2023.1249142

Salami et al. 2024. Dissection of quantitative trait nucleotides and candidate genes associated with agronomic and yield-related traits under drought stress in rapeseed varieties: integration of genome-wide association study and transcriptomic analysis. Frontiers in Plant Sciences, 15 doi.org/10.3389/fpls.2024.1342359

Archangi et al. 2022. Assessing genetic diversity and aggregate genotype selection in a collection of cumin (Cuminum cyminum L.) accessions under drought stress: Application of BLUP and BLUE. Scientia Horticulturem 299, 11108.. //doi.org/10.1016/j.scienta.2022.111028

Archangi et al. 2019. Association between seed yield-related traits and cDNA-AFLP markers in cumin (Cuminum cyminum) under drought and irrigation regimes. Industrial Crops and Products, 133: 276-283. //doi.org/10.1016/j.indcrop.2019.03.038

Results

-Line 312: the authors stated 62 plant samples that was not consistent with the number of accessions explained in materials and methods. Overall, materials and methods are in some places confusing as the authors did not explain the experiments and number of plant samples used for different assays

-line 346: Mantel not Mental test

-line 351: one of reasons for identifying low number of linked SNP is low plant sample size used for GWAS analysis

Discussion

Comparing with other studies is poor and discussion part need more literature review and avoid stating detailed results. Fosus on main and key finding and interpret the results. Revise discussion part as suggested.

Conclusion

Conclusion is too large and should be condensed intro 2-3 sentences stating the most important finding not explain everything

Figures and tables:

Resolution of figures are too low. It is not suitable for publication

Reviewer #2: I have gone through the manuscript “Morphological characteristics and genome-wide association analysis among local Andrographis paniculata from Thailand under controlled environment in plant factory” focusing on the identification of high yielding variety based on the content of andrographolide and biomass under Plant factories with artificial lighting (PFAL) of different cultivars. I feel it’s an excellent work done by the authors because the lant which has been selected is widely used in all over the world and the objective taken in the study is really need of the hour because in the change in environment the medicinal properties may also have changed in the plants. However, I have very few suggestions on the manuscript to improve the quality and readability of the paper which are as follows.

1. What the numeral ‘1’ in the tittle. It can be removed

2. Instead of writing only ‘Andrographis’ in many places, in should be A. paniculata in all the laces where it is mentioned.

3. The morphological description has revealed the leaves are typically 2-12 cm long, this data should be rechecked. The citation should be given from where the description has been verified (Any flora book or reference book or research paper).

4. In line number 61, the ‘2’ should be in the subscript of carbon dioxide.

5. In line number 172, check the spelling of A. paniculata

6. The most important question which need to be answered in the manuscript is, phylogenetically the accessions like CR, RB, PL, and PC—are closely related to TTT but even though provided with similar conditions the TTT given higher yield. Please explain what could be the reason in the conclusion part.

6. PLOS authors have the option to publish the peer review history of their article (what does this mean? ). If published, this will include your full peer review and any attached files.

**Do you want your identity to be public for this peer review?** For information about this choice, including consent withdrawal, please see our Privacy Policy .

Reviewer #1: No

Reviewer #2: **Yes: ** Bibhuti Bhushan Champati

---

## [Author Response · Author response to Decision Letter 1]

9 Feb 2025

Responses to Reviewers’ Comments

Response: Based on the PLOS ONE formatting and file naming guidelines, we revised following PLOS ONE guidelines.

2. Thank you for stating the following financial disclosure: [This research was funded by the support of National Science and Technology Development Agency, Thailand (P2351505) and Thailand Basic Research Fund: fiscal year 2023 with Contract no. 4709540.]. Please state what role the funders took in the study. If the funders had no role, please state: "The funders had no role in study design, data collection and analysis, decision to publish, or preparation of the manuscript." If this statement is not correct you must amend it as needed. Please include this amended Role of Funder statement in your cover letter; we will change the online submission form on your behalf.

Response: We have clarified the role of our funders in the revised cover letter. The National Science and Technology Development Agency, Thailand (P2351505) and the Thailand Basic Research Fund (Contract no. 4709540), primarily provided financial support for the research. They did not have any role in the study design, data collection and analysis, decision to publish, or preparation of the manuscript. This statement will be included in our cover letter.

Response: I have ensured that my ORCID iD is linked and validated in the Editorial Manager. This was done through the 'Update my Information' section, where I used the Fetch/Validate link to confirm my ORCID iD. This will facilitate compliance with PLOS's requirements for author identification.

Response: We have amended the title to ensure that it is identical in both the online submission form and the manuscript itself.

5. Review Comments to the Author

Reviewer #1: 1-As one of important parts of the paper is about GWAS analysis, it is suggested to address high quality publications about the merits of GWAS analysis in different species in including medicinal plants. Bellow, several publications about GWAs and meta-analysis can be addressed in introduction as literature review and in discussion part for comparing similar results between the publications and the submitted manuscript.

Shariatipur et al. 2021. Comparative Genomic Analysis of Quantitative Trait Loci Associated With Micronutrient Contents, Grain Quality, and Agronomic Traits in Wheat (Triticum aestivum L.). frontiers in Plant Science, //doi.org/10.3389/fpls.2021.709817

Shariatipur et al 2021. Meta-analysis of QTLome for grain zinc and iron contents in wheat (Triticum aestivum L.). Eyphytica 217, //doi.org/10.1007/s10681-021-02818-8

Shariatipour et al. 2021. Genomic analysis of ionome-related QTLs in Arabidopsis thaliana. Scientific Reports, 11. doi.org/10.1038/s41598-021-98592-7

Salami et al. 2022. Comparative profiling of polyphenols and antioxidants and analysis of antiglycation activities in rapeseed (Brassica napus L.) under different moisture regimes. Food Chemistry, 399: //doi.org/10.1016/j.foodchem.2022.133946

Salami et al. 2023. Integration of genome wide association studies (GWAS), metabolomics and transcriptomics reveals phenolic acids and flavonoids associated genes and their regulatory elements under drought stress in rapeseed flowers. Frontiers in Plant Science, 14, 10.3389/fpls.2023.1249142

Salami et al. 2024. Dissection of quantitative trait nucleotides and candidate genes associated with agronomic and yield-related traits under drought stress in rapeseed varieties: integration of genome-wide association study and transcriptomic analysis. Frontiers in Plant Sciences, 15 doi.org/10.3389/fpls.2024.1342359

Archangi et al. 2022. Assessing genetic diversity and aggregate genotype selection in a collection of cumin (Cuminum cyminum L.) accessions under drought stress: Application of BLUP and BLUE. Scientia Horticulturem 299, 11108.. //doi.org/10.1016/j.scienta.2022.111028

Archangi et al. 2019. Association between seed yield-related traits and cDNA-AFLP markers in cumin (Cuminum cyminum) under drought and irrigation regimes. Industrial Crops and Products, 133: 276-283. //doi.org/10.1016/j.indcrop.2019.03.038

Response: Thank you for your feedback regarding our manuscript. We appreciate your suggestions on enhancing our discussion of GWAS analysis in different species, including medicinal plants. We have added more high-quality GWAS studies to our introduction. These studies demonstrate the application of GWAS in various species, providing a solid foundation for our research. (Introduction section)

In the discussion section, we now compare our findings with those in the newly added references. This helps to clarify how our results fit into the broader scientific context. We believe these updates have made our manuscript stronger and more informative. Thank you for helping us improve our work. (Discussion sections)

Results

-Line 312: the authors stated 62 plant samples that was not consistent with the number of accessions explained in materials and methods. Overall, materials and methods are in some places confusing as the authors did not explain the experiments and number of plant samples used for different assays.

Response: Thank you for pointing out the inconsistency regarding the number of plant samples discussed in the Results and Materials and Methods sections. We acknowledge the confusion and appreciate the opportunity to clarify this aspect of our study.

In the Materials and Methods section, we initially reported planting seeds from ten different Andrographis accessions. For the genome-wide association studies (GWAS) and other genetic analyses, we expanded the number of samples to include multiple plants per accession to ensure robust genetic analysis. Specifically, we used a total of 62 individual plants representing a mix of the ten accessions, with the number of plants per accession varying based on the specific experimental needs for genetic diversity and replication.

To better align the sections of the manuscript and eliminate any confusion, we have revised the corresponding sentences in “Sample collection and RADseq sequencing” of Materials and Methods section.

-line 346: Mantel not Mental test-line 351: one of reasons for identifying low number of linked SNP is low plant sample size used for GWAS analysis

Response: Thank you for your meticulous review and for pointing out the errors. Line 346 has corrected the typo from "Mental" to "Mantel" to accurately refer to the Mantel test, which is used for assessing the correlation between distance matrices.

Line 351: Sample size is one of the possible factors related to the low number of detected significant SNP. However, it is also related to the number of genetic loci controlling the target traits. GWAS analysis will show the SNP position that the majority of accessions associated to the trait and reduce error of detection by factor of each GWAS model. If the sample size is small and the traits are controlled by multiple loci (or different genes that exhibit the same phenotype, such as disease resistance genes), there is a low likelihood that the associated SNPs will pass the significance cutoff. However, if the small sample size includes enough accessions to represent the type of SNP or has common loci controlling the trait, significant SNPs can still be identified. Moreover, the low number of linked SNPs may also depend on the number of SNPs used in the analysis, which is related to the SNP filtering conditions.

Discussion

Comparing with other studies is poor and discussion part need more literature review and avoid stating detailed results. Fosus on main and key finding and interpret the results. Revise discussion part as suggested.

Response: We appreciate your constructive feedback regarding the Discussion section of our manuscript. We reduced the detailed presentation of individual results and instead emphasized a more integrated comparison with other relevant studies. Moreover, we incorporated additional literature that further contextualizes our findings within the existing body of research, particularly focusing on how our results align with or diverge from those studies. (Discussion section)

Conclusion

Conclusion is too large and should be condensed intro 2-3 sentences stating the most important finding not explain everything

Response: Thank you for your comments on the conclusion section of our manuscript. We have condensed this section into a more succinct summary, focusing only on the most critical findings of our research. The revised conclusion now succinctly states the primary outcomes and their significance in two sentences, ensuring clarity and brevity as suggested. (Conclusion section)

Figures and tables:Resolution of figures are too low. It is not suitable for publication

Response: Thank you for your feedback concerning the resolution of the figures and tables included in our manuscript. We have thoroughly revised all figures and tables to ensure they meet publication standards. To address the issue, we utilized PACE (Publication-quality Artwork Conversion Engine), which enhances image resolution and ensures each figure is optimized for both digital and print formats.

Reviewer #2: I have gone through the manuscript “Morphological characteristics and genome-wide association analysis among local Andrographis paniculata from Thailand under controlled environment in plant factory” focusing on the identification of high yielding variety based on the content of andrographolide and biomass under Plant factories with artificial lighting (PFAL) of different cultivars. I feel it’s an excellent work done by the authors because the lant which has been selected is widely used in all over the world and the objective taken in the study is really need of the hour because in the change in environment the medicinal properties may also have changed in the plants. However, I have very few suggestions on the manuscript to improve the quality and readability of the paper which are as follows.

1. What the numeral ‘1’ in the tittle. It can be removed

Response: We have removed the numeral ‘1’ from the title as it was indeed unnecessary and could cause confusion.

2. Instead of writing only ‘Andrographis’ in many places, in should be A. paniculata in all the laces where it is mentioned.

Response: We have revised the text to consistently use the scientific name “A. paniculata” instead of the common name “Andrographis” throughout the document to maintain scientific accuracy and consistency.

3. The morphological description has revealed the leaves are typically 2-12 cm long, this data should be rechecked. The citation should be given from where the description has been verified (Any flora book or reference book or research paper).

Response: We have rechecked the morphological data concerning the leaf length, which is mentioned as typically ranging from 2-12 cm. This measurement has been verified with data sourced from (Hossain et a., 2014), which we have now cited in the manuscript to substantiate our descriptions. (Line 34: Introduction section)

Reference

Hossain MS, Urbi Z, Sule A, Hafizur Rahman KM. Andrographis paniculata (Burm. f.) Wall.

ex Nees: a review of ethnobotany, phytochemistry, and pharmacology. ScientificWorldJournal. 2014;2014:274905. doi:10.1155/2014/274905.

4. In line number 61, the ‘2’ should be in the subscript of carbon dioxide.

Response: In line 61, we have corrected the notation of carbon dioxide to include the subscript in ‘CO₂’ to adhere to scientific standards.

5. In line number 172, check the spelling of A. paniculata

Response: We have reviewed and corrected the spelling of "A. paniculata" in line 172 and ensured that it is consistently correct throughout the manuscript.

6. The most important question which need to be answered in the manuscript is, phylogenetically the accessions like CR, RB, PL, and PC—are closely related to TTT but even though provided with similar conditions the TTT given higher yield. Please explain what could be the reason in the conclusion part.

Response: The phylogenetic analysis showed that CR, RB, PL, PC, and TTT were closely related. The samples grouped in the same clade or cluster can exhibit similarities or differences in phenotype; for instance, in this study, TTT yielded a higher output. This indicates that certain parts of their genomes differ. The phylogenetic analysis was conducted using 16,431 variant calls, representing the number of variants remaining after filtering, which reflects some variance in their genomes.

---

## [Decision Letter · Decision Letter 1]

23 Feb 2025

Morphological characteristics and genome-wide association analysis among local Andrographis paniculata from Thailand under controlled environment in plant factory

PONE-D-24-60023R1

Dear Dr. Chutimanukul,

We’re pleased to inform you that your manuscript has been judged scientifically suitable for publication and will be formally accepted for publication once it meets all outstanding technical requirements.

Kind regards,

Faham Khamesipour, Ph.D.

Academic Editor

PLOS ONE

Additional Editor Comments (optional):

The authors have adressed all comments systematically. I agree to the changes made by the authors.

Reviewers' comments:

Reviewer's Responses to Questions

**Comments to the Author**

1. If the authors have adequately addressed your comments raised in a previous round of review and you feel that this manuscript is now acceptable for publication, you may indicate that here to bypass the “Comments to the Author” section, enter your conflict of interest statement in the “Confidential to Editor” section, and submit your "Accept" recommendation.

Reviewer #1: All comments have been addressed

Reviewer #2: All comments have been addressed

2. Is the manuscript technically sound, and do the data support the conclusions?

Reviewer #1: Yes

Reviewer #2: Yes

3. Has the statistical analysis been performed appropriately and rigorously? 

Reviewer #1: Yes

Reviewer #2: Yes

4. Have the authors made all data underlying the findings in their manuscript fully available?

Reviewer #1: Yes

Reviewer #2: Yes

5. Is the manuscript presented in an intelligible fashion and written in standard English?

Reviewer #1: Yes

Reviewer #2: Yes

6. Review Comments to the Author

Reviewer #1: The comments addressed properly and this version of the manuscript is acceptable for publication. Juts check the text for possible typo errors

Reviewer #2: (No Response)

7. PLOS authors have the option to publish the peer review history of their article (what does this mean? ). If published, this will include your full peer review and any attached files.

**Do you want your identity to be public for this peer review?** For information about this choice, including consent withdrawal, please see our Privacy Policy .

Reviewer #1: No

Reviewer #2: **Yes: ** Bibhuti Bhushan Champati

---

## [Editor Report · Acceptance letter]

PONE-D-24-60023R1

PLOS ONE

Dear Dr. Chutimanukul,

I'm pleased to inform you that your manuscript has been deemed suitable for publication in PLOS ONE. Congratulations! Your manuscript is now being handed over to our production team.

Kind regards,

on behalf of

Dr. Faham Khamesipour

Academic Editor

PLOS ONE